# Ventral tegmental area dopamine projections to the hippocampus trigger long-term potentiation and contextual learning

**Fares J. P. Sayegh** [1] ✉, **Lionel Mouledous** [1], **Catherine Macri** [1], **Juliana Pi Macedo** [1], **Camille Lejards**[1], **Claire Rampon** [1], **Laure Verret** [1] & **Lionel Dahan** [1] ✉

In most models of neuronal plasticity and memory, dopamine is thought to promote the long-term maintenance of Long-Term Potentiation (LTP) underlying memory processes, but not the initiation of plasticity or new information storage. Here, we used optogenetic manipulation of midbrain dopamine neurons in male DAT::Cre mice, and discovered that stimulating the Schaffer collaterals – the glutamatergic axons connecting CA3 and CA1 regions - of the dorsal hippocampus concomitantly with midbrain dopamine terminals within a 200 millisecond time-window triggers LTP at glutamatergic synapses. Moreover, we showed that the stimulation of this dopaminergic pathway facilitates contextual learning in awake behaving mice, while its inhibition hinders it. Thus, activation of midbrain dopamine can operate as a teaching signal that triggers NeoHebbian LTP and promotes supervised learning.

In an attempt to explain associative learning, Donald Hebb's proposed that "when an axon of cell A (...) repeatedly or persistently takes part in firing of cell B, some growth process or metabolic change takes place in one or both cells such that A's efficiency, as one of the cells firing B, is increased"[1]. Experimental support for Hebb's rule came through the discovery of Long-Term Potentiation (LTP): an activity-dependent enhancement of synaptic transmission able in some cases to persist for many hours. Although LTP has been demonstrated at glutamatergic synapses in most regions of the mammalian brain, it has been studied most extensively in the CA1 region of the hippocampus[2]. LTP can be triggered in neuronal culture, in slices, or in vivo by a strong stimulation of the presynaptic axons. Convenient protocols for triggering LTP include high frequency stimulation (HFS), typically a train of 50-100 stimuli at 100 Hz, or theta burst stimulations (TBS), consisting in several bursts of 4 to 6 shocks at 100 Hz with an interburst interval of 200 ms[2,3]. It can also be triggered by a spike-timing-dependent plasticity (STDP) protocol[4], in which the post-synaptic cell is repeatedly forced to fire an action potential 1 to 20 milliseconds after the presynaptic cell. Here it gives the appearance that the presynaptic element is participating in the activity of the postsynaptic cell, as postulated by Hebb.

These artificial protocols have enabled the mechanisms of synaptic plasticity to be deciphered. The induction of LTP relies on the N-Methyl-D-Aspartate (NMDA) receptor, described as a coincidence detector, that allows calcium influx in the postsynaptic element when glutamate is released while the postsynaptic element is in a depolarized state[5]. This calcium influx activates kinases, including Calmodulin-dependent protein Kinase II (CaMKII) and other kinases such as Protein Kinase A (PKA), Protein Kinase C (PKC) or mitogen-activated protein kinases (MAPK), depending on the experimental conditions. By phosphorylating α-amino-3-hydroxy-5-methyl-4-isoxazolepropionic acid (AMPA) receptors, their biophysical properties are changed, and their number at the synapse increased. These processes result in an increased level of glutamatergic transmission that can last for a few hours, a phenomenon that has been called early-LTP[2]. If the triggering stimulus is sufficiently strong, kinases will additionally trigger protein synthesis that will allow morphological changes making LTP last longer, several hours, even days; this is called late-LTP[6]. Finally, learning itself can also trigger synaptic plasticity resembling hippocampal LTP in rats[7] or mice[8,9]. Learning-induced plasticity appears to share common mechanisms with LTP triggered by HFS, TBS and STDP,

[1]Centre de Recherches sur la Cognition Animale (CRCA), Centre de Biologie Intégrative (CBI), Université de Toulouse; CNRS, UPS, Toulouse, France.
✉e-mail: fares-sayegh@hotmail.com; lionel.dahan@univ-tlse3.fr

thereby suggesting the same phenomenon can be triggered both by artificial and natural protocols. Since the same cellular mechanisms are at play, Hebbian LTP is considered one of the main mechanisms underlying hippocampus-dependent learning and memory[10].

However, the Hebbian learning rules have been largely limited to unsupervised learning tasks[11], which in turn would suggest that a hippocampus relying purely on Hebbian rules should encode every single event. However, as elegantly expressed by Frémaux and Gerstner[12], "After exposure to cars passing by on a highway, we do not remember every single one, but most often only a few relevant cars, such as the most salient, novel, or surprising items". This means that a filter is required to ensure that only important items are remembered, thereby preventing constant overwriting of old memory and avoiding overloading the computational capacity of the hippocampus. This led theoreticians to propose a NeoHebbian framework in which the conjunction between pre- and post-synaptic activity requires a third factor - a neuromodulator playing the role of a teaching signal–in order to trigger long-lasting synaptic plasticity[13,14].

Midbrain dopamine neurons located in the ventral tegmental area (VTA) emerge as leading candidates for providing the teaching signal to the hippocampus. They fire in response to salient events, including rewards[15], aversive[16] and novel stimuli[17,18]. In other words, they fire whenever something is worth remembering. Although the hippocampus is not a major efferent target, recent studies show that a subset of VTA dopamine neurons innervate the dorsal hippocampus, and in particular CA1 pyramidal layer[9,19]. Furthermore, pharmacological studies have established the role of dopamine receptors in the hippocampal LTP[20,21] (for reviews[22,23]). Thus, a majority of studies report that antagonists of dopamine D1/5 receptors block late-LTP[20,24] and agonists are able to turn early-LTP induced by a weak HFS into a long-lasting late-LTP[20]. Thus, it is usually considered that dopamine is required for making LTP and memory persist[25]. However, the role of dopamine in synaptic plasticity might be more nuanced. The administration of dopamine antagonists (i.c.v.) prevents the novelty-induced facilitation of LTP induction in vivo[26]. In vitro, dopamine broadens the temporal window for potentiation and increases its amplitude when applied to the perfusion bath during the induction of STDP[27] or during the 10 minutes following the pairings[28]. Most importantly, in vitro bath application of dopamine agonist is able to trigger a slow onset LTP resembling late-LTP[29], while in vivo intraperitoneal administration was shown to completely block learning-induced LTP[9]. Thus, depending on the experimental protocol, dopamine might play a role in triggering hippocampal LTP or in its transition to protein synthesis-dependent late-LTP. However, prior pharmacological approaches have two major drawbacks: (i) they are anatomically non-specific therefore prevent the study of a particular pathway; and (ii) the relative timing of the dopamine input and glutamate releases required for triggering LTP cannot be investigated.

The question of the anatomical pathway is important because noradrenergic neurons of the locus coeruleus (LC) massively innervate the hippocampus and seem able to release dopamine which activates D1/5 receptors therein[18]. Since this LC-hippocampus pathway can also modulate hippocampal LTP and learning[18,30], the pharmacology experiments cannot distinguish an involvement of the VTA or LC innervation. The only available study investigating the modulation of glutamatergic transmission by specific stimulation of the VTA-hippocampus pathway reported a short-term potentiation (15–30 minutes) but did not assess any further longer-lasting effects compatible with LTP[31]. The purpose of the present study was therefore to investigate specifically the involvement of VTA dopaminergic neurons projections in hippocampal LTP.

Within the framework of NeoHebbian plasticity, the pattern of activation of the glutamatergic synapse and its timing relative to dopaminergic release is also of major importance. Dopamine could induce LTP through two potential scenarios: first, by being released concomitantly with synaptic activation to allow novelty-based learning; and second, by being released with a delay of a few seconds to allow reward-based learning[13]. The second timeframe would correspond with the delay between a rewarded event and the actual delivery of the reward[12]. In an in vitro experiment, Yagishita and colleagues used precisely timed optogenetic stimulation of dopamine neurons in acute slices of the Nucleus Accumbens to show that dopamine promoted spine enlargement - thought to reflect late-LTP - only during a narrow time window (0.3 to 2 seconds) after the beginning of a STDP[32]. Unfortunately, the STDP protocol lasted for 1 second, which makes the precise timing required between the activation of the glutamatergic synapse and the dopamine signal difficult to interpret. From these data, one might conclude that dopamine must be released during the pre/postsynaptic pairings, or during 1 second immediately following the pairings. A more recent publication shows that cortico-striatal synapses can be potentiated in vivo if dopamine afferents are repeatedly activated 250 ms to 1 second after a single pulse stimulation of the glutamatergic input[33]. Thus, in the striatum, dopamine must be released after the activation of the glutamatergic synapse, within a short time window compatible with reward-based instrumental learning. Consequently, a second aim of the current study was to determine the effects of differently timed dopamine release in relation to single pulse stimulation of glutamatergic synapses in the hippocampus.

In summary, the first part of this study sought to confirm the ability of the VTA-hippocampus dopamine pathway to initiate LTP and to pinpoint the time window during which dopamine can fulfil this role. We therefore conducted a series of experiments in anesthetized mice where extracellular electrodes were used to stimulate and monitor the local field potentials as a measure of synaptic transmission at Schaffer collaterals, and optogenetic tools were used to specifically activate VTA dopamine nerve terminals in the hippocampus.

Dopamine antagonists were reported to spare the early phase of LTP while blocking the late maintenance of LTP requiring protein synthesis[9,10,20] and synaptic tagging[6]. Consequently, behavioral research has focused mainly on the role of dopamine mediating novelty-induced enhancement of memory and behavioral tagging[34]. Based on the electrophysiological findings in the first part of this study showing that VTA dopamine can promote hippocampal LTP, and on a previous pharmacological study[35], we investigated the possibility that midbrain dopamine neurons are involved in learning a new context. In said study, we employed a modified version of contextual fear conditioning known as the Context Pre-exposure Facilitation Effect (CPFE). In this procedure, mice learn separately a context in the absence of reinforcement, then its association with an electric shock; pre-exposure to the context facilitates subsequent fear conditioning. Using this paradigm, we found that the facilitation effect was blocked when dopamine D1/D5 receptor antagonists were administered during the context pre-exposure. Although this study highlighted the role of dopamine in contextual learning, we were unable to differentiate between the effects of blocking dopamine signals originating from the LC or the VTA. Additionally, since the antagonistic effects persisted beyond the learning phase, we couldn't rule out the possibility of the pharmacological treatment affecting the consolidation process rather than memory acquisition itself.

Thus, in the second part of this study, we tested whether the VTA-hippocampus dopamine pathway can promote novelty-based learning in the absence of reward. We used the CPFE procedure and optogenetically stimulated or inhibited the terminals of VTA dopamine neurons innervating the dorsal hippocampus during the period when the animals were pre-exposed to the context.

## Results

We optogenetically manipulated VTA midbrain dopamine neurons and tested whether these inputs to the hippocampus trigger LTP. We injected a viral vector expressing the ultra-fast excitatory ChETA[36]

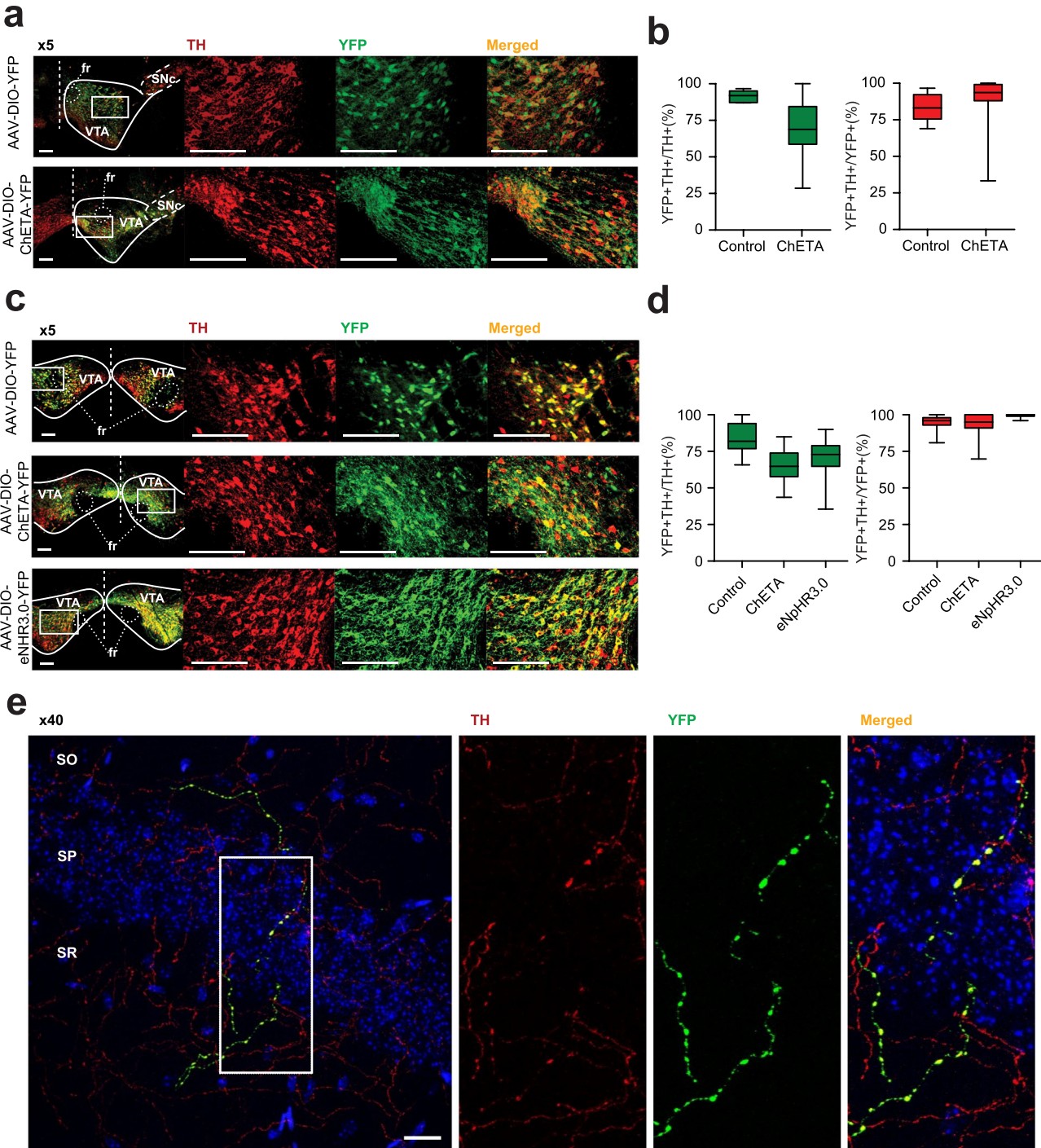

**Fig. 1 | Specific transfection of the VTA-hippocampus dopamine pathway.**
**a** Transfected coronal sections by unilateral injection for electrophysiology recording stained for Tyrosine Hydroxylase (TH) and YFP. **b** quantification for unilaterally injected mice. Transfection and specificity (transfection, specificity) were (91.5 ± 2%, 83.4 ± 4%, $n = 6$ mice) for YFP and (70 ± 2%, 91.5 ± 2%, $n = 57$ mice) for ChETA injected mice. **c** Transfected coronal sections by bilateral injection for behavioral experiments stained for TH and YFP. d. Quantification for transfection and specificity (transfection, specificity) were (83.9 ± 2%, 94.7 ± 1%, $n = 27$ mice) for YFP, (65.9 ± 2%, 93.3 ± 2%, $n = 34$ mice) for ChETA and (70.8 ± 5%, 99.4%, $n = 11$ mice) for eNpHR3.0 bilaterally injected mice. For panels **a** & **c,** Dashed line represents midline; rectangle represents sampled area captured at x20 magnification. fr, fasciculus retroflexus; SNc, Substantia Nigra pars compacta; VTA, Ventral Tegmental

Area. Scale bars, 100 μm. For panels **b** and **d** Quantification of Transfection (green) and Specificity (red) are represented as the number of double-stained cells divided by TH+ cells and YFP+ cells, respectively. In the box plots representations, whiskers show the minimum and maximum values, bounds of the box the 1st and 3rd quartile and center line indicate the median. **e** Representative image of confocal microscopy examination of coronal sections of the dorsal hippocampus of mice injected bilaterally with the YFP virus ($n = 27$) which revealed a sparse labeling in the CA1 region that was not restricted to a specific layer. Dopaminergic fibers originating from the VTA and expressing YFP are shown in green. TH containing fibers are in red, while cell nuclei labeled with Hoechst are in blue. SO, stratum oriens; SP, stratum pyramidale; SR, stratum radiatum. Scale bar, 20 μm. Source data are provided as a Source Data file.

rhodopsin (AAV2-Ef1a-DIO-ChETA-EYFP), or its EYFP control (AAV2-Ef1a-DIO-EYFP), into the VTA of DAT::Cre[37] mice. Both vectors induced efficient and specific transfection of dopamine neurons in the VTA and medial Substantia Nigra pars compacta (Fig. 1a, b). We observed sparse YFP labeling in the CA1 region that was not restricted to a specific layer (Fig. 1e). These YFP-labeled fibers co-express TH, as would be expected for axons arising from the transfected dopamine neurons in the VTA. Most TH fibers showed no YFP signal, which is consistent with a labelling of a mix of axons arising from a few non transfected fibers, and a majority of fibers arising from noradrenergic neurons from the Locus Coeruleus. In urethane anesthetized mice, electrophysiological recordings were used to measure evoked field potentials at Schaffer collaterals in vivo. An optical fiber was placed into the glass recording pipette for the purpose of optically stimulating midbrain dopamine terminals located in recorded area of CA1 (Fig. 2a).

Our main finding was that fifty pairings of one light burst (4 ms pulses at 50 Hz during 400 ms) delivered concomitantly with single pulse electrical stimulations of Schaffer collaterals (0.1 ms, delivered 200 ms after the onset of the light burst) induced an increase in the slope of Schaffer collaterals' field potentials (+55 ± 14%) (Fig. 2b, ChETA Paired). Importantly, this synaptic plasticity developed progressively over 90 minutes and remained stable for at least 5 hours after the last coupling. We called this phenomenon DA-LTP. This protocol failed to induce plasticity in mice transfected with the control vector (1.7 ± 1.8%) (Fig. 2b, YFP Paired), or in mice transfected with ChETA vector but receiving optogenetic stimulations unpaired with the electrical stimulation of the Schaffer collaterals (−2.1 ± 5%) (Fig. 2b, ChETA Unpaired group, in which the same number of optogenetic stimulations and the same number of Schaffer collaterals stimulations were delivered as in the ChETA paired group; the only difference being that in the ChETA unpaired group there was a delay of 15 + \− 2.5 seconds between the optical and the electrical stimulations). DA-LTP was prevented by intraperitoneal injection of D1/5 R antagonist SCH23390 (0.05 mg/kg) which did not alter synaptic transmission (+1.1 ± 3% for SCH23390 injected mice vs. +42 ± 10% for saline injected mice) (Fig. 2c).

LTP is a saturable process[38]. This property can be used to demonstrate that two plasticity processes, for example, learning-induced LTP and HFS-LTP, share some mechanisms when one occludes the other[7]. We thus conducted an occlusion experiment to investigate whether DA-LTP and classical LTP share similar mechanisms. The first experimental group, referred to as 'DA-LTP', received a ChETA Paired protocol that induced a significant LTP (Fig. 2d, in blue). The second group, referred to as 'no DA-LTP', received either ChETA Unpaired or YFP Paired protocols that did not induce any synaptic plasticity (Fig. 2d, in grey). After a 90-minute interval, both groups received TBS, and fEPSPs were re-normalized, using the 50 minutes preceding the TBS as baseline 2. Notably, both groups exhibited similar levels of post-tetanic potentiation, a short-term plasticity typically induced by TBS and independent of LTP mechanisms[39,40]. Conversely, the subsequent LTP was maintained at a lower magnitude in the DA-LTP group (23.9 ± 5%) compared to the No DA-LTP group (47.2 ± 6%) (Fig. 2d). Therefore, DA-LTP partially occluded LTP triggered by TBS. In view of this result, we suspect the presence of a shared underlying mechanism governing the expression and maintenance of DA-LTP and TBS-LTP.

To determine the time window within which the pairing was required to trigger DA-LTP, we reduced light bursts duration from 400 ms to 200 ms, which also better mimics a natural burst of dopamine[41,42]. DA-LTP was triggered when dopamine-terminal stimulation was delivered simultaneously with the electrical stimulation of Schaffer collaterals and for the next 200 ms (0 to 200 ms, +46 ± 20). No DA-LTP occurred when the dopamine stimulation occurred prior to the Schaffer stimulation (−200 to 0 ms, -6 ± 5%), nor when it was delayed by more than 200 ms after the electrical stimulation (+200 to +400 ms, -3.2 ± 12%) (Fig. 2e). By progressively reducing the number of

pairings, we found that 12 pairings were sufficient to induce a full DA-LTP (+46.5 ± 16%), while 6 pairings had no lasting effect (+3.3 ± 2%) (Fig. 2f). Hence, triggering DA-LTP requires 7 to 12 dopamine stimulations mimicking naturally occurring bursts. These results demonstrate that repeated bursts of activity of midbrain dopamine afferents to the hippocampus can effectively trigger LTP at coactivated Schaffer collaterals.

To test whether VTA dopamine input to the hippocampus is also important for hippocampus-dependent learning we used the Context Pre-exposure Facilitation Effect (CPFE) protocol. In this procedure, mice were first exposed to an environmental context, and then on the following day its association with an electric shock. On the third day, conditioned fear was assessed with measures of behavioral freezing (Fig. 3a). We first validated this CPFE protocol in DAT::Cre mice by showing that sufficient pre-exposure to the context was necessary for the facilitation effect to be observed. Thus, both control mice (not pre-exposed to the context), and mice pre-exposed for 30 seconds exhibited low levels of freezing on Day 3 (37 ± 6% and 42.1 ± 5%, respectively, Fig. 3b). Conversely, in animals pre-exposed for 2 or 8 min, to allow contextual learning, the amount of freezing observed on Day 3 was significantly increased (62.7 ± 4% and 61.7 ± 5%, Fig. 2b). We then optogenetically manipulated dopamine input to the hippocampus during the pre-exposure session to determine its role in contextual learning in the CPFE protocol.

DAT::Cre mice received a bilateral injections of either the control vector (AAV2-Ef1a-DIO-EYFP), or a vector expressing ChETA (AAV2-Ef1a-DIO-ChETA-EYFP) or eNpHR3.0 (AAV2-Ef1a-DIO-eNpHR3.0-EYFP) in the VTA. All three vectors induced efficient and specific transfection of dopamine neurons in the VTA and medial Substantia Nigra pars compacta (Fig. 1c, d). To manipulate the VTA-hippocampus pathway specifically, optical fibers were inserted bilaterally over the dorsal CA1.

First, optogenetically released dopamine into the hippocampus during the short 30 second pre-exposure period enabled the pre-exposure facilitation effect. Mice expressing ChETA in VTA dopamine neurons and receiving 90 bursts (burst duration: 200 ms, 4 ms pulses, @50 Hz) of blue light (473 nm, 10 mW) exhibited significantly higher levels of freezing on Day 3 (62.8 ± 4%) than that observed in YFP mice (45.6 ± 3%; Fig. 3c). Alternatively, mice expressing the inhibitory opsin eNpHR3.0 in dopamine VTA neurons and pre-exposed to the context for 2 minutes while receiving a continuous green light (532 nm, 10 mW, starting 20 s before placing the mouse in the context) in the hippocampus showed low levels of freezing (47.2 ± 6%), which was significantly lower than that of mice receiving the same treatment but injected with the YFP control vector (79.2 ± 6%). The level of freezing observed in the eNpHR3.0 group was comparable to that observed with non-pre-exposed mice, showing that the facilitation effect induced by the 2 minutes pre-exposure was entirely blocked by the optogenetic inhibition of dopamine terminals. In both experiments, freezing in an alternative context was considerably lower than in the conditioned context, and was not modified by dopaminergic activation or inhibition during pre-exposure (Figs. 3c, d). Thus, midbrain dopamine inputs to the dorsal hippocampus provide a signal triggering learning regardless of value inputs.

## Discussion

This study has identified conditions in which dopamine projection from the VTA to the dorsal hippocampus was able to trigger hippocampal LTP in vivo and provides evidence that this pathway contributes to novel contextual learning.

The dorsal hippocampus is not a major output target of midbrain dopamine neurons[43] and previous reports utilizing axon-targeted channelrhodopsin viral methodologies[18,30,44] have suggested that hippocampal innervation by these neurons is sparse. Using the same methodology, we also confirmed this sparse labelling of transfected fibers in the dorsal hippocampus. Consequently, the discovery that

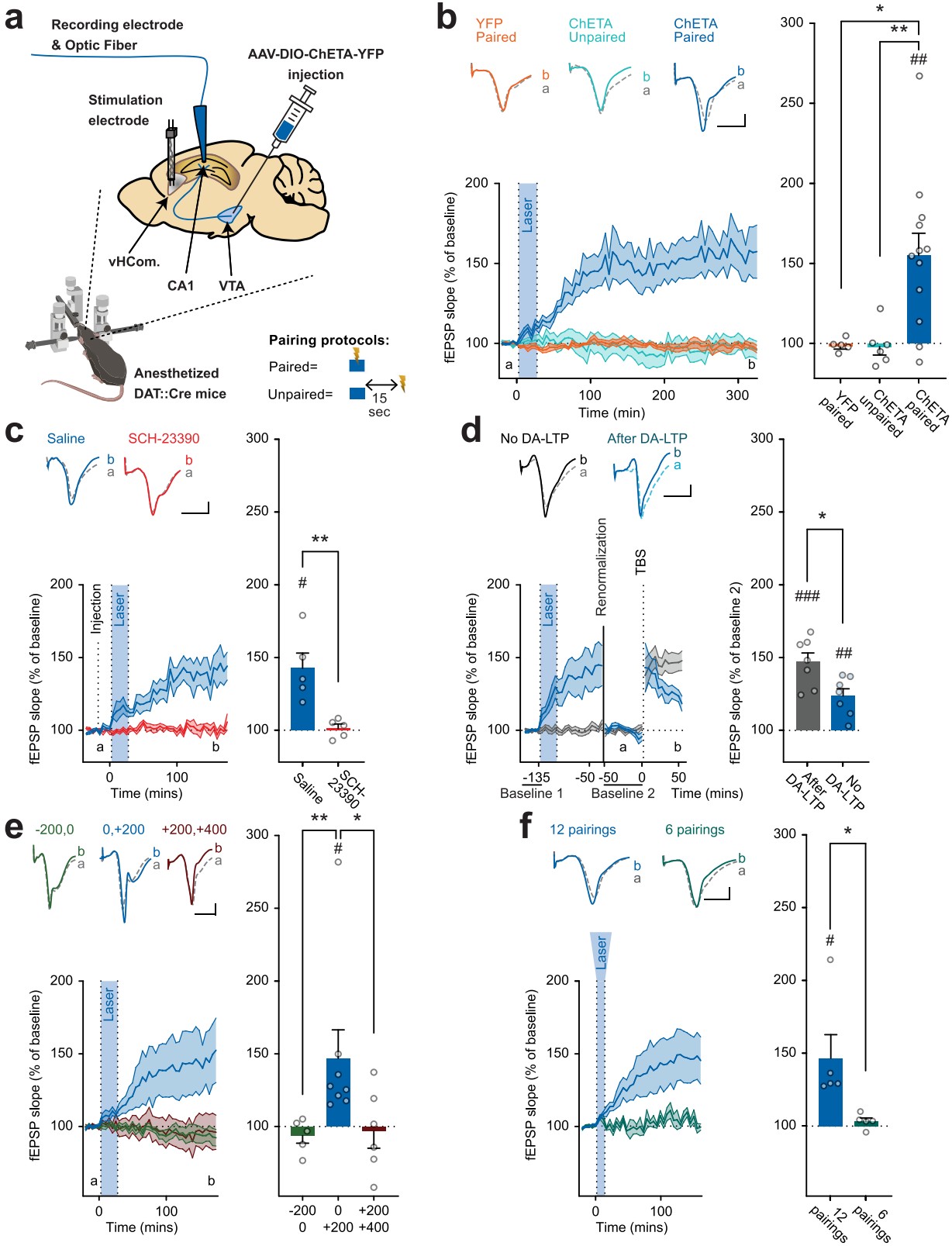

optogenetic manipulation of the noradrenergic neurons of the LC modulates hippocampal synaptic transmission and memory through D1/5 receptors[18,30] reignited the debate regarding the source of dopaminergic innervation of the hippocampus and questioned the relevance of the VTA-hippocampus dopamine pathway to hippocampal function. Thus, the robust effects we observed in both our electrophysiological and behavioral experiments may seem surprising.

However, a recent study using anterograde and retrograde viral tracing showed that a cluster of midbrain dopamine neurons located in the lateral VTA provides a dense innervation of the pyramidal cell layer of CA1[19]. Subsequent optogenetic stimulation of VTA dopamine terminals in the hippocampus was shown to modulate fear conditioning. In agreement with Tsetsenis and colleagues[19], and without questioning the role of dopaminergic transmission originating from

**Fig. 2 | Midbrain dopamine triggers a long-lasting, D1/5R-dependent, increase of synaptic transmission in CA1, which occludes TBS-triggered LTP in DAT::Cre mice. a** a schema representing the procedure. DAT::Cre mice were injected with either floxed YFP-coding vectors or ChETA and YFP-coding vectors in their Ventral Tegmental Area (VTA). Three weeks later, we performed anesthetized in vivo electrophysiology recording of CA1 response to Schaffer collaterals electric stimulation in the ventral hippocampal commissure (vHCom.) every 30 seconds. Optic stimulation was delivered through the glass recording pipette following paired or unpaired protocols (lightning represents electrical stimulations and blue rectangle represents one light burst of 4 ms pulses @50 Hz during 400 ms of laser stimulation). Illustration include an image created with BioRender.com. **b** Effect of 50 coupling optical stimulations and electrical stimulations of Schaffer Collaterals (blue shaded part of the timeline). When stimulations were simultaneously coupled in ChETA injected mice, an increase in fEPSP slopes was observed (Dark Blue, ChETA paired, t-test vs 100%: $p = 0.0021$, $n = 12$ mice). No such increase was observed in control vector injected mice (Orange, YFP paired, t-test vs 100%: $p = 0.38$, $n = 5$ mice) nor when electrical and optogenetic stimulations were separated by 15 seconds (Light Blue, ChETA Unpaired, t-test vs 100%: p = 0.70, $n = 6$ mice). The increase in the ChETA paired group is statistically different from other groups (Kruskal Wallis: p = 0.01, Mann-Whitney post hoc tests: $p = 0.79$ for ChETA unpaired vs. YFP, $p = 0.0097$ for ChETA paired vs. ChETA unpaired and $p = 0.02$ for ChETA paired vs. YFP). **c** SCH23390 injected 20 minutes prior to the coupling (dashed line); EPSP slope increase was no longer observed in the SCH23390 group (red, t-test vs 100%: $p = 0.72$, $n = 5$ mice) while significant in the control group (dark blue, t-test vs 100%: $p = 0.015$, $n = 5$ mice). The difference between the 2 groups is

statistically significant (Mann-Whitney test: $p = 0.0079$). **d** Classic form of LTP was induced 90 minutes after the end of couplings using TBS. DA-LTP group (in blue) showed a rapid degradation of TBS induced LTP. Both groups show a significant LTP induced by TBS (t-test vs 100%: $p = 0.0003$ for NoDA-LTP (in grey) and $p = 0.0032$ for DA-LTP, $n = 7$ mice for each group). The difference between the 2 groups is statistically significant (Mann-Whitney test: $p = 0.026$). **e** DA-LTP was induced when optical stimulations were delivered 0 to 200 ms after the electrical stimulation (dark blue, t-test vs 100%: $p = 0.048$, $n = 8$ mice), but not when it was were delivered 200 to 0 ms before (dark green) nor 200 to 400 ms after (dark red) the electrical stimulation (t-test vs 100%: $p = 0.29$ and 0.79 and $n = 5$ and 6, respectively). The increase in the 0; + 200 ms group is statistically different from other groups (Kruskal Wallis: p: 0.0054, Mann-Whitney post hoc tests: $p = 0.0016$ for −200;0 vs. 0;+ 200, $p = 0.043$ for 0;+ 200 vs. +200;+ 400 and $p = 0.99$ for −200;0 vs. +200;+ 400). **f** 12 pairings of optical stimulations (0 to 200 ms in relation to SC electrical stimulations) was sufficient to induce DA-LTP (dark blue, t-test vs 100%: $p = 0.044$, $n = 5$ mice), but not 6 (light green, t-test vs 100%: $p = 0.21$, $n = 5$ mice). The difference between the 2 groups is statistically significant (Mann-Whithney test: $p = 0.0079$). For panels b to f, timelines of each group on the left, mean changes quantified by averaging the last 25 minutes of the recording for each mouse on the right. Data are presented as mean values +/- SEM. Sample size (n) indicates the number of mice included for each experimental group. * $p < 0.05$ Mann-Whitney, ** $p < 0.01$ Mann-Whitney (after significant Kruskal Wallis). # $p < 0.05$ t-test vs. 100%. ## $p < 0.01$ t-test vs. 100%. ### $p < 0.001$ t-test vs 100%. All statistical tests were two-sided. Source data are provided as a Source Data file.

the LC, the present study further confirms specific physiological and behavioral functions of the VTA-dorsal hippocampus dopamine pathway.

As detailed in the introduction, the role of dopamine in hippocampal LTP is often limited to the modulation of LTP triggered by HFS or STDP protocols. Dopamine antagonists administered after HFS shortens the maintenance of LTP to less than an hour and a half[20,22]. Dopamine was therefore presumed to participate in the initiation of the synthesis of proteins required for Late-LTP[45]. In brain slices, the application of dopamine during the STDP pairing protocol facilitates and widens the time window during which STDP pairing is effective[27,28,46]. The experimental conditions used to demonstrate these effects were very different from ours. We did not apply any HFS or STDP protocols, rather, we simply stimulated the Schaffer collaterals repeatedly, every 30 seconds, to induce the synaptic transmission on to which we superimposed optogenetic stimulation of afferent VTA dopamine terminals for 12 to 50 cycles. This procedure was sufficient to trigger an LTP that lasted for at least 5 hours. At this stage, we cannot with certainty rule out that the postsynaptic elements may have responded to the stimulation of the presynaptic elements by firing action potentials. The shape of the recorded field potentials varied and, in some cases, an inflexion compatible with a population spike was sometimes observable (see Fig. 2 for field potential examples). However, in the absence of optogenetic dopamine stimulation, electrical stimulation of the Schaffer collaterals failed to trigger any plasticity, contrary to what Hebb's postulate would have predicted if the stimulation of the presynaptic elements were to repeatedly induce postsynaptic action potentials. Subsequent in vitro patch clamp experiments will be necessary to determine how far optical stimulation of dopamine terminals coupled with the firing of action potentials in the postsynaptic element might further enhance DA-LTP, or whether its coupling with synaptic transmission evoking only EPSPs is sufficient. In any case, our findings demonstrate that neither HFS nor STDP induction protocols are required for triggering hippocampal LTP; the repeated concomitant activation of Schaffer collaterals and VTA dopamine terminals is sufficient for this effect.

In the striatum or in the Nucleus Accumbens, dopamine is able to trigger potentiation at corticostriatal synapses on medium spiny neurons (MSNs) that were depolarized 200 ms to 2 seconds before dopamine[32,33,47]. This delay, is supposed to allow computing an

eligibility trace, enabling the striatum to bridge the gaps between an action and the unpredicted sensory consequences it causes[13] in order to deal with motor learning[48] and value processing[49]. Attempts to show eligibility traces in the hippocampus are not as convincing as in the striatum. Using in vitro recordings, Brzosko and collaborators were able to turn the LTD triggered by a post-before-pre (−20ms) STDP protocol into an LTP by applying dopamine to the perfusion bath within a minute following the STDP protocol[28]. Although it shows that dopamine might modulate previously occurring plasticity events, both the STDP protocol and the DA application used in this study lasts 10 minutes, which is not ideal to assess time windows compatible with eligibility traces. These results are probably better related to the concept of synaptic consolidation rather than to eligibility trace[13]. Since, in this work, the effective time window for hippocampal dopamine to trigger DA-LTP is restricted to the 200 ms following glutamate release, it is probable that midbrain dopamine afferents to the hippocampus are not involved in computing eligibility trace. This may reflect differences in the functions performed by the basal ganglia and the hippocampus. The hippocampus plays a role in episodic-like memories and the requirement for synchronous glutamatergic and dopaminergic signals are perfectly suited for novelty- or surprise-based learning, as proposed by Gerstner and collaborators[13].

In each of our experiments, DA-LTP slowly develops over 60 to 90 min before reaching a plateau. In that sense, it lacks any sign of early-LTP or post-tetanus potentiation seen in most experiments studying HFS-triggered LTP[20,50]. Our temporal curve resembles D1/5 receptor-triggered LTP observed by Navakkode and collaborators[29,51]. In rat hippocampal slices they reported that the application of the D1/5 receptor agonists SKF38393 or 6-bromo-APB for 30 minutes triggered LTP, but only if glutamatergic inputs were stimulated by low-frequency single pulse test stimulations during the D1/D5-receptor activation[29,51]. This D1/5-LTP was also blocked by NMDA Antagonist AP5, inhibitors of CamKII and MAPK/ERK Kinases and protein synthesis inhibitors. This shows that the coactivation of D1/5 receptors and glutamatergic transmission was necessary to trigger this LTP, which was considered late-LTP in the absence of any early-LTP. Given their similarities, we propose that the DA-LTP observed in this study and D1/5-LTP observed by Navakkode and collaborators might be one same mechanism. Here we show that DA-LTP occurs in vivo, relies on inputs arising from

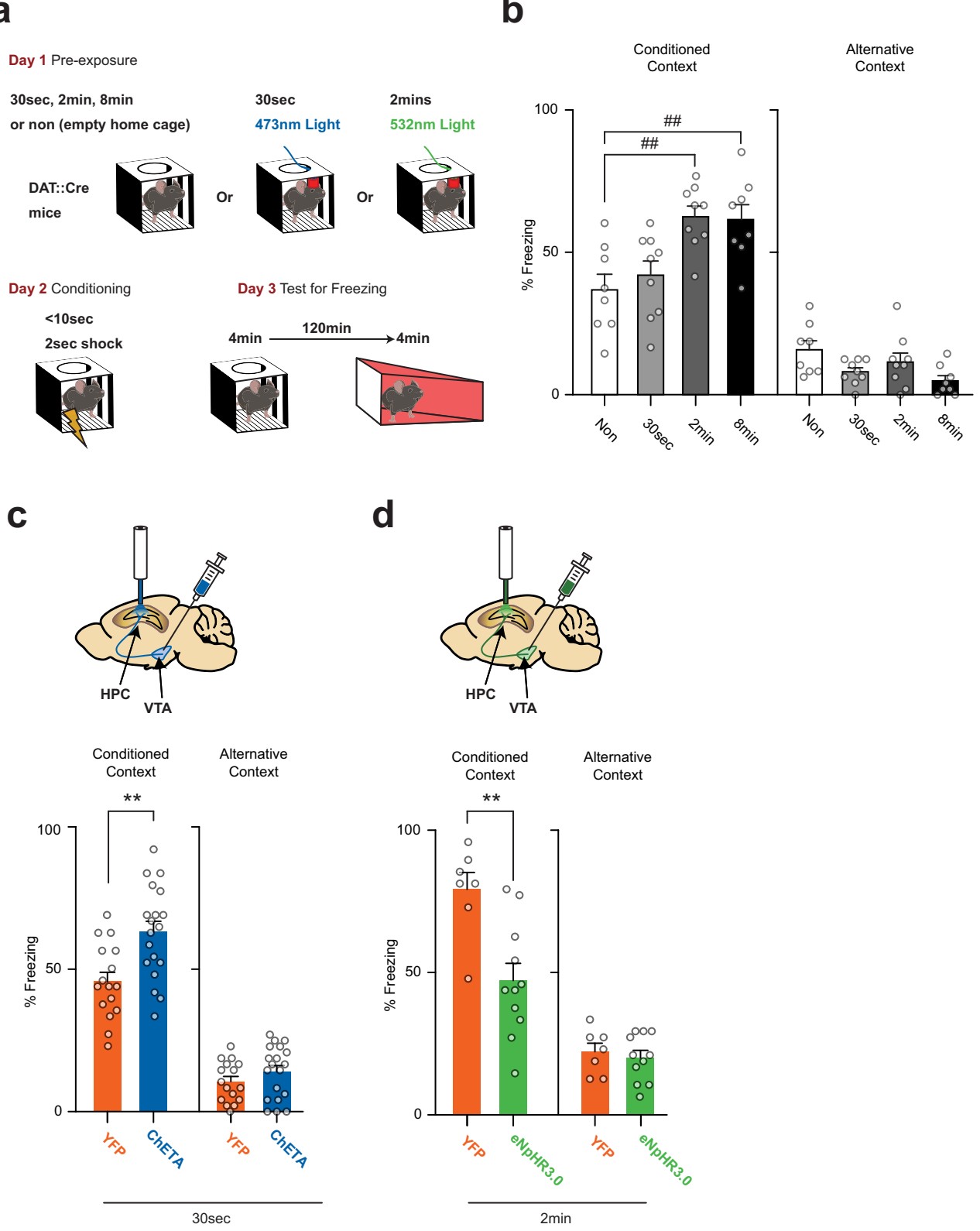

midbrain dopamine neurons, and can be triggered by precisely timed bursts of dopamine close to those observed in natural circumstances.

Translated in natural conditions, our electrophysiological results showing DA-LTP would mean that when dopamine cells innervating the hippocampus fire bursts of action potentials, the Schaffer Collaterals synapses that are concomitantly activated – at a 200 ms time scale – would be potentiated. In this regard, the requirement of 7 to 12 pairings to trigger LTP could allow filtering only the set of synapses that relates to unpredicted stimuli which had triggered the dopamine burst and was reliably concomitant. These characteristics of midbrain dopamine inputs to the hippocampus correspond to the definition of the teaching signal previously hypothesized to trigger NeoHebbian LTP involving a third factor – here dopamine – in response to rewarding, aversive or neutral unpredicted events[13,23].

**Fig. 3 | Midbrain dopamine in the hippocampus contributes in learning a new context in DAT::Cre mice. a** Schema illustrating the behavioral procedure. DAT::Cre mice were used in a context pre-exposure facilitation effect paradigm. First, the effect of different pre-exposure durations was studied, and then the effect of optogenetic manipulation of midbrain dopamine afferents to the hippocampus was evaluated. On day 2, mice received an immediate shock, and on day 3 freezing was tested in the conditioned context and in an alternative one. **b** Preexposure had a significant effect on freezing to the conditioned context (ANOVA, $p = 0.0008$) in DAT::Cre mice. Animals pre-exposed for 30 seconds ($n = 9$ mice) did not freeze more than control non-pre-exposed group ($n = 8$ mice, Tukey post hoc test: $p = 0.78$). 2 min pre-exposure was sufficient to induce a significant increase ($n = 9$ mice, Tukey post hoc test: $p = .0024$) reaching levels comparable to those seen in the group with 8 min pre-exposure ($n = 8$ mice, Tukey post hoc test: $p = 0.0041$). **c** Mice were pre-exposed for 30 seconds on day one during which they received 90 bursts of blue light bilaterally in the dorsal hippocampus. Freezing in the conditioned context increased in the ChETA-injected mice (blue, $n = 19$ mice) compared

the YFP-injected mice (orange, $n = 16$ mice, t-test: $p = 0.0016$). Freezing in the alternative context was considerably lower than in the conditioned context and was not significantly changed due to dopaminergic activation during pre-exposure (t-test: $p = 0.23$). **d** Mice were pre-exposed for 2 minutes on day one during which they received continuous green light. Freezing in the conditioned context levels observed during the test on day 3 were lower for eNpHR3.0 injected mice (green, $n = 11$ mice) in comparison to mice with control injection (orange, $n = 7$ mice, t-test: $p = 0.0021$). Freezing in the alternative context was considerably lower than in the conditioned context and was not significantly changed due to dopaminergic inhibition during pre-exposure (t-test: $p = 0.60$). Data are presented as mean values +/- SEM. Sample size (n) indicates the number of mice included for each experimental group. ## $p < 0.01$ Multiple comparisons following one-way ANOVA ($p < 0.001$). ** $p < 0.01$ t-test. All statistical tests were two-sided. Illustrations include an image created with BioRender.com. Source data are provided as a Source Data file.

We show that the stimulation of VTA dopamine terminals in the hippocampus can trigger the encoding of contextual memory while their inhibition can completely prevent it. This corroborates that midbrain dopamine neurons provide a teaching signal to the hippocampus. Dopamine has already been associated with learning tasks relying on the hippocampus. Broussard and collaborators showed that learning-induced hippocampal LTP, presenting a slow development similar to DA-LTP, is inhibited by the D1/5 antagonist SCH23390[9]. However, it is worth noting that the intraperitoneal administration of D1/5 receptors may have indirect effects on other brain structures, potentially affecting motivation and attention, which could explain the observed effects without necessarily requiring dopamine to trigger LTP in the hippocampus. More recently, the same team showed that specifically activating midbrain dopamine afferents to the hippocampus with optogenetic tools concomitantly with shock delivery in a fear conditioning paradigm facilitates fear memory encoding[19]. This result can be interpreted as an enhancement of memory for the context, or for the shock, or for the association of both stimuli. We obtained very similar results using a similar stimulation protocol during the pre-exposure session of the CPFE when the animals were freely exploring the environment. Thus, we further show that hippocampal dopamine innervation can trigger learning a new context in the absence of any aversive or rewarding event. Additionally, the major contribution of our behavioral study is to provide evidence that specifically inhibiting the dopaminergic pathway from the midbrain to the hippocampus prevents contextual learning. In line with our electrophysiological results, these data show that VTA dopamine afferents to the hippocampus are involved in triggering the latent encoding of new contexts.

Our findings may seem inconsistent with two influential reports in the literature proposing it is dopamine released from LC, and not VTA, that facilitates memory[18,30]. On the one hand, it is important to keep in mind the significant experimental differences between our study and the one reported by Takeuchi et al.[18]. They used the everyday spatial memory task, in which mice have to learn the location of a reward in a well-known event arena. In this test, mice show a preference for the rewarded location one hour after the learning session but not 24 h later, unless they were exposed to a novel environment 30 minutes after the learning session. The memory-promoting effect of exposure to novelty was dependent on D1/5 receptors, mimicked by the optogenetic activation of the LC and not blocked by the inactivation of the VTA by local lidocaine infusion. Thus, this experiment shows that activation of D1/5 receptors in the hippocampus by neurotransmitters released by LC terminals is involved in behavioral tagging and capture[18]. However, this study did not assess the role of dopamine during the learning session itself, as we did here. On the other hand, Kempadoo *et al.* showed that the optogenetic stimulation of the LC terminals in the hippocampus during the acquisition of an object

localization task improved memory, an effect that was prevented by D1/5 but not by β-adrenergic antagonists[30]. Given that they demonstrated a significant release of dopamine in the slices of dorsal hippocampus when they optogenetically stimulate the LC terminals, they argued that LC is the predominant dopaminergic influence in this region. However, they did not assess the effect of manipulating VTA terminals and therefore this study is not in contradiction with the results reported here.

Noteworthily, dopamine released by the LC terminals in the hippocampus was also recently shown to play a role in linking the memory for different contexts, but not in contextual learning per se[52]. Thus, dopamine probably contributes to memory in different ways depending whether it is released in the hippocampus by terminals arising from the LC or the VTA. These different contributions might be due to different localization of dopamine synapses within the hippocampus. Indeed, VTA terminals are confined to the pyramidal layer of the CA1 area, while LC terminals are more widespread within all layers of the whole hippocampus[19]. Since we show that VTA projections to the hippocampus are both sufficient and necessary for contextual learning, we propose that dopamine released in the hippocampus by VTA terminals is involved in learning new contexts and forging new memories while dopamine released from the LC would play a role in updating and linking previously established memories.

In summary, our in vivo electrophysiological experiments in anaesthetized mice showed that optical activation of afferent midbrain dopamine projections to the dorsal CA1 region of the hippocampus triggered DA-LTP. This effect was mediated though D1/5 receptors when dopamine was released in conjunction with electrical activation of Schaffer collaterals. In behavioral experiments, we show that similar optogenetic activation of afferent midbrain dopamine terminals in the dorsal hippocampus promoted contextual learning, while inhibition hindered it. We propose that by triggering hippocampal LTP in novel circumstances, dopamine may act as a teaching signal for encoding relevant sensory inputs to be memorized.

## Methods
### Animals
Mice 2-6.5-month-old male DAT::Cre[37] mice were bred in-house. DAT::Cre mice are descendants of FVB/N with Cre-recombinase under the control of the dopamine transporter promoter (DAT) inserted into a bacterial artificial chromosome (BAC) then mice were backcrossed more than 15 times on the C57BL/6 J line. These animals are placed in stalls in groups of 3 to 5 individuals per cage, with food and water available ad libitum, under a 12 h/12 h day/night cycle (day from 8 h to 20 h) at $22 \pm 1\,°C$ and 50% humidity. All experiments were performed in strict accordance with the recommendations of the European Union (2010/63/EU) and the French National Committee (2013-118) under the

guidance of the local ethical committee for animal experimentation of the Federation de Recherche en Biologie de Toulouse (FRBT).

## Electrophysiology

Anesthesia was induced using Isoflurane then maintained with an injection of urethane ($1.5 \pm 0.2$ mg/g) during the recording period. In the stereotaxic apparatus mice received an incision and a craniotomy, then stimulating electrodes were placed at the ventral hippocampal commissure (Coordinates: -0.3 mm AP, -0.5 mm ML and 2.3 DV) to evoke field excitatory postsynaptic potential (fEPSP) recorded at the stratum radiatum of area CA1 recorded using a glass micro-pipette (NaCl 2 M filled 4microns thick 0.8-1.1 Megaohm @100 Hz). Recordings were performed using an A-M Systems model 1800 amplifier (gain = 100, bandwidth of 0.1 Hz to 10 kHz) and stimulations using 2100 Isolated Pulse Stimulator from A-M Systems both connected to CED micro 1401 which allowed sampling at 10 kHz after 50 Hz filtration using Humbug 50 Hz Noise Eliminator, QuestScientific. Electrical stimulations were delivered to the Schaffer collaterals every 30 seconds, with a jitter of 5 seconds (the interval between two consecutive stimulations ranged from 27.5 to 32.5 seconds). Signals were visualized and analyzed using Spike2. The raw electrophysiological signal was first analyzed by averaging fEPSP waveforms every 5 minutes (10 fEPSP). The slope of the initial phase of each mean fEPSP is measured. A baseline was obtained by stimulating with a current providing 70% of the maximal response recorded (0.05-1 mA biphasic 100μs stimulations), the same stimulation was maintained throughout the recording including the plasticity-induction protocols. We considered 25 min of recording (i.e., five mean fEPSPs) with a slope between 95 and 105% of the average as a stable baseline and the evolution of the fEPSP slope should not follow a linear change. We then delivered the coupling of glutamatergic stimulation and dopaminergic optical stimulation, after which a follow up took place. For the occlusion experiment, mice received a HFS protocol of LTP. This protocol followed Theta Burst Stimulation (TBS) pattern, for which we used 4 trains (30 s inter-train interval) of 4 bursts (200 ms inter-burst interval) of 5 stimulations at 100 Hz. At the end of the recording, the glass pipette is replaced with one filled with Chicago blue 2% in acetate buffer 0.5 M. We searched for the same recorded fEPSP and the recording site was marked using electric expulsion of the dye with negative current of 20 μA, cycles 10 seconds "on" / 10 seconds "off" for 10 minutes. Finally, the mice were euthanized with a lethal injection of pentobarbital, and then, received an intracardial infusion of 0.9% NaCl solution and the brains were collected for immunohistochemical verification.

## Vectors

For optogenetics manipulations we used UNC Vector Core provided vectors; AAV2-Ef1a-DIO-ChETA-EYFP, AAV2-Ef1a-DIO-EYFP or AAV2-Ef1a-DIO-eNpHR3.0-EYFP at original concentrations at purchase ($3.5$-$5 \times 10^{12}$ particle/mL). For electrophysiological experiments, mice received 2 unilateral injections of 0.5 μL/Site at ($-3.3$ mm AP, $+0.6$ mm ML and $-4.0/-4.6$ mm DV). For behavioral experiments mice received 2 bilateral injections 0.3 μL/Site at ($-3.3$ mm AP, $\pm0.6$ mm ML and $-4.0/-4.6$ mm DV).

## Laser delivery

Two lasers were used for this work 473 nm DPSS Laserglow (blue light for ChETA activation) and 532 nm DPSS Laserglow (green light for eNpHR3.0 activation). Light was then connected to the head of the mouse either through 200 μm patch chords, then through 200 μm nude optical fibers and finally, through the implanted cannulas (for behavioral studies). For electrophysiology, longer optical fiber cannulas were placed inside the recording glass micro-pipette that led the light to the recording site. The intensity was set to 10 mW at the implantable tip. For inhibition, mice with eNpHR3.0 vectors received one continuous pulse (140 seconds long) covering 20 seconds before

context pre-exposure the 2 minutes of pre-exposure. For stimulation, mice received either 200 ms or 400 ms bursts (4 ms pulses, 50 Hz) depending on the protocol. These pulses were driven using Model 2100 Isolated Pulse Stimulator from A-M Systems and Spike2 connected through Micro1401 from CED.

## Behavioral task

For Context Pre-exposure Facilitation Effect (CPFE), on day 1, mice were allowed to explore the context (27x27x27cm, with horizontal black and white stripes on a wall and vertical ones on the opposite, metal rods floor, white light) during 30 seconds, 2 minutes or 8 minutes. We used as control a non-pre-exposed group that explored an empty home cage for 8 minutes. On day 2, all animals received a very brief conditioning session ( < 10 seconds) during which they received one electric shock (0.7 mA, 2 s) in the conditioned context. Tests took place on day 3; freezing behavior was measured in the same context for 4 minutes to assess contextual memory. An hour and a half later, freezing was measured in an alternative context (triangular shape, transparent Plexiglas walls and floor, red light) for 4 minutes to assess generalized fear, mice showing >33.33% generalization were excluded from the analysis (1 from non-pre-exposed mice and 4 YFP). Freezing was scored each 5 seconds by two independent experimenters blind to the experimental condition and expressed as a percentage of the sampled time spent freezing. In order to respect parametric requirements for ANOVA tests, percentages (P) were transformed using the equation $Q = A\sin(\sqrt{P/100})$. To make sure that odor association did not happen, we used nonalcoholic makeup remover wipes (Ysiance, aloe vera) to clean the contexts on the first two days and alcohol solution on day three. The experimenter who conducted the conditioning session on day 2 was different from the experimenter on days 1 and 3 in order to avoid any association between the context, the experimenter and the shock. Before optogenetic manipulations, mice were first habituated to the connection of the optical fibers during the week preceding the experiments by thrice connecting them to patch cords and allowing them to explore an empty home cage once every two days. Mice injected with either ChETA coding vector or its control were pre-exposed to the context for 30 seconds during which they received 90 bursts of blue light bilaterally in the dorsal hippocampus (burst duration: 200 ms, 4 ms pulses @50 Hz, 473 nm, 10 mW). The optical stimulation protocol was chosen to match the activity of dopamine neurons we previously recorded during REM sleep of palatable food consumption[41]. Two other groups, injected with either eNpHR3.0 coding vector or its control, and were pre-exposed to the context for 2 minutes during which they received a continuous green light bilateral illumination of the hippocampus (532 nm, 10 mW).

## Immunohistochemistry

To measure the efficiency of the transfection, we carried out a double immunohistochemical labeling revealing the reporter protein eYFP, and Tyrosine Hydroxylase (TH) the enzyme necessary for dopamine production, to visualize the transfected cells and dopaminergic cells, respectively. The animals were anesthetized (pentobarbital) before performing an intracardiac infusion with 0.9% NaCl solution (20-30 s, 20 mL/min). The brains were then removed and placed in 4% PFA solution for 24-72 hr, then rinsed with 0.1 M PBS. Finally, the brains were stored in a 30% sucrose solution containing 0.1% azide. These brains were later sectioned into several serial 40 μm sections with a freezing microtome, then stored in a cryoprotectant solution. For immunohistochemistry staining, sections were washed in 0.1 M phosphate buffered saline containing 0.25% triton (PBST), placed for 15 min in a solution containing 3% $H_2O_2$ and 10% methanol (in PBST), to block the endogenous peroxidase. After two rinses of 10 min with PBST, sections were placed for 1 hour in a solution saturating non-specific bonds (5% donkey serum in PBST). Finally, they were incubated overnight at room temperature in a solution of 5% donkey serum in PBST

containing the primary antibodies goat anti-YFP (1:2,500; Rockland, 600101215) and rabbit anti-TH 1:1,000 (Millipore, AB152). The next day, the sections were twice rinsed in PBST before being placed for 90 minutes in a solution containing the fluorescent secondary antibodies (donkey anti-goat A488 1:250 (Thermofisher, A11055) and donkey anti-rabbit A555 1:250 (Thermofisher, A31572)). Finally, the sections were twice rinsed in PBST before mounting them on Superfrost slide. Slide covers were glued with Mowiol containing Hoechst (1: 10,000) in order to mark nuclei. Once dry, the slides were observed using a Leica DM6000 B fluorescence microscope, a sampled transfection zone was counted for each mouse (using Mercator software, Explora Nova). The GFP and TH antibodies have been validated for Immunofluorescence by the manufacturer and used in several publications referenced on the manufacturer's website at the following address (https://www.merckmillipore.com/FR/fr/product/Anti-Tyrosine-Hydroxylase-Antibody,MM_NF-AB152 and https://www.rockland.com/categories/primary-antibodies/gfp-antibody-600-101-215/).

Sections were photographed using the same software and these photos were retouched using ImageJ. Slides were then stored at 4 °C. Transfection analysis relied on two metrics, transfection and specificity by studying the collocation of TH expressing cells and YFP expressing cells.

Specificity was calculated by the equation:

$$specificity\% = \frac{\#[TH+,YFP+]}{\#[YFP+]} \times 100$$

Transfection was calculated by the equation:

$$transfection\% = \frac{\#[TH+,YFP+]}{\#[TH+]} \times 100$$

Confocal images of the CA1 region of the hippocampus were acquired using a Leica TCS SP8 microscope (Heidelberg, Germany) equipped with 40x oil objective (software: LasX 1.4.5, Leica microsystems). 8-bit images were taken in z-series over 20 µm at a resolution of 1024 ×1024 pixels. ImageJ (version 1.53t) was then used to obtain a 2D image using the Maximum Intensity Z-projection function.

### Data analysis
Power calculations were employed to determine the appropriate number of animals for each group, with a predetermined a priori statistical power set at 90% (using Minitab). All statistical tests and figures were done using GraphPad Prism 8 and graphs were retouched using Adobe Illustrator CS6. fEPSP slopes were analyzed using the tools provided in spike2 and freezing was analyzed offline by two independent experimenters blind to the experimental conditions.

### Reporting summary
Further information on research design is available in the Nature Portfolio Reporting Summary linked to this article.

## Data availability
For each figure, source data are provided as a Source Data file. All raw data are available for all readers without any restriction on demand by mail to lionel.dahan@univ-tlse3.fr. Source data are provided with this paper.

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

## Acknowledgements

We would like to thank Dr. Stephanie Trouche for her valuable insights into the behavioral experiments, Drs. Emmanuel Valjent and Dina Arvanitis for their wise advice throughout the project, and Professors Christian Luscher, Richard Morris, and Peter Redrave for their invaluable feedback on our initial manuscript. This work did not receive any specific grant funding and was fully supported by the recurrent funding of the CRCA by CNRS and Université Paul Sabatier – Toulouse. Illustrations include images created with BioRender.com.

## Author contributions

F.J.P.S Conceptualization, Formal Analysis, Investigation (performed most of the histological, electrophysiological and behavioral experiments), Methodology, Visualization, Writing – original draft, Writing – review & editing. L.M.: Investigation (performed confocal analysis of the histological preparations), Visualization, Writing – review & editing. C.M.: Investigation (participated to behavioural experiments). J.P.M.: Investigation (participated to behavioural experiments). C.L.: Investigation (participated to the histological preparations). C.R.: Resources, Writing – review & editing. L.V.: Resources, Writing – review & editing. L.D.: Conceptualization, Formal Analysis, Funding acquisition, Investigation (participated in electrophysiological and behavioral experiments), Methodology, Project administration, Supervision, Visualization, Writing – original draft, Writing – review & editing.

## Competing interests

All authors declare no competing interests.
