## [Peer Review File · Nature Communications]

Ventral Tegmental Area dopamine projections to the hippocampus trigger long term potentiation and contextual learningREVIEWER COMMENTS

Reviewer #1 (Remarks to the Author):

In this brief manuscript the authors investigated the pairing of single pulses of Schaffer-collateral electrical stimulation with optogenetic burst stimulation of VTA dopamine terminals, on the strength of CA1 hippocampal fEPSPs. They investigate (a) the effect of varying the relative timing of activation of these inputs, (b) the number of pairings of afferent activation with dopamine release, (c) the effect of manipulating dopamine D1-receptor activation on plasticity, and (d) the behavioral effect of DA burst firing or inhibition at the stage of learning a context, prior to learning foot-shock association in that context.

The strengths of this study are: (i) that it is *in vivo*, (the use of anesthesia is always an issue that most slice physiologists push back on, however I am not one!), (ii) that the authors show convincing immunohistochemistry that demonstrates sufficient and specific transduction of DA cells in DAT-Cre mice, (iii) they use appropriate inactive laser and control-vector controls, (iv) they attempt to clarify the importance of relative timing of afferent inputs to plasticity effects, and (v) they undertake separate behavioral experiments that aim to give 'context' to the plasticity effects, by demonstrating that dopamine burst stimulation in awake behaving mice enhances contextual learning (by increasing freezing behavior to a previous foot-shock-paired environment), whereas inhibiting dopamine firing decreases contextual learning to control levels.

For the most part, the study is well-performed and convincing, but the weaknesses include (i) lack of detail of some methodological steps, (ii) issues with interpretation of some of the experiments, and (iii) the degree of novelty of the study in addressing the role of dopamine in hippocampal plasticity induction. The authors should consider the following:

1) The introductory paragraph gives a very abbreviated description of the background to hippocampal LTP (the m/s was I assume submitted as a brief communication but this was not clear from the website which classifies it as an "Article" – even so, the background and justification should be a more thorough). I am sure the authors are aware that there are many forms of induction protocol for hippocampal LTP, many that have had extensive work detailing the cellular induction mechanisms. This background needs expansion, summarising briefly the different forms of induction protocol and focussing this down onto the particular form for which this paper seeks to reveal a novel mechanism.

2) The plasticity induction protocols used for each experiment are buried within the text of the legends and only briefly described – there needs a figure inset illustrating exactly the pairing protocol used and the relative pairing timings used in experiments that varied these. What intensity was the electrical stimulation set to that was applied during the plasticity-induction protocol (I realise that fEPSPs were elicited at 70% max response but how was this varied during the induction protocol and what was the post-synaptic effect)? Was there evidence of population spike induction during the protocol or was the intensity subthreshold? Was there a fibre volley visible and if so how did it change? What was the interval between each of the 50 pairings (i.e. interval from electrical-laser to electrical-laser stimulation)?

3) What was the detail of the optogenetic activation of dopamine fibres that was unpaired with Schaffer collateral stimulation? Were the same number of Schaffer-collateral stimulations given but randomised in relation to the optogenetic activations or was the electrical stimulation not given at all? Was the optogenetic activation given as the same number of activations over a similar total period as the paired experiments (assuming that these applications were randomised in comparison to electrical stimulations). This is a very important experiment since if the same number of stims as opto activations are given but one of them is randomly applied in relation to the other it would speak more to the plasticity induced during pairing still as a Hebbian mechanism with the addition of a temporally-specific third factor (dopamine) as a neuromodulator (see Izhikevich, E.M. Solving the distal reward

problem through linkage of STDP and dopamine signaling), rather than, as the authors refer to it, a non-Hebbian induction protocol.

4) The justification that the authors give for the novelty of their study states that “no study addressed whether and how midbrain dopamine may trigger hippocampal LTP[14], which is the missing part for fully understanding dopamine's role in allowing supervised learning outside the Hebbian framework[15]”. I would challenge this with the findings of Brzosko, Z. et al (2015), who showed that a post-pre pairing at a very short interval (-10ms) was able to induce LTP due to release of endogenous dopamine. Their plasticity and its timing dependence was able to be blocked by bath-applied D1 antagonist SCH23390. Hence, although the present study used HFS and measured fEPSP slope, there still would have been a conjunction of pre- and post activity coupled with a ‘third factor’ involving dopamine release working at D1 receptors. This study should be considered in the context of the current findings as evidence of DA D1 involvement in the induction of hippocampal plasticity.

5) Fig. 1e is given as evidence that plasticity induced by their DA-LTP pairing protocol occludes subsequent plasticity induction by another protocol TBS, suggesting a common mechanism. Notably, there was an induction of DA-LTP first, then a readjustment of stimulus intensities to baseline (pre potentiation) levels and then the application of TBS. They describe the result as a partial occlusion, however the immediate magnitude of potentiation induced by TBS maps on exactly to control levels; it is just the persistence which is reduced, albeit a change still there in an hour. A true occlusion of mechanism should not have produced any further increase in fEPSP after baseline levels were readjusted. I think this is weak evidence that should be enhanced or qualified with further experiments/analyses or substituted with another experiment seeking mechanism.

6) Minor – there are quite a lot of typos/tense and plural issues that will need correcting.

References

1. Izhikevich, E.M. Solving the distal reward problem through linkage of STDP and dopamine signaling. *Cereb Cortex* 17: 2443-52 (2007).
2. Brzosko, Z., Schultz, W., & Paulsen, O. Retroactive modulation of spike timing-dependent plasticity by dopamine. *Elife* 4 (2015).

Reviewer #2 (Remarks to the Author):

The authors demonstrate that dopamine projections from the ventral tegmental area (VTA) can initiate long term potentiation (LTP) in the dorsal hippocampus of anesthetized mice. This is shown using repeated in vivo electrical stimulation of the Schaffer collaterals with optogenetic stimulation of midbrain dopamine afferents to CA1. The successful induction of hippocampal DA-LTP involves D1/5 receptors and enables mice to acquire contextual fear conditioning. Authors conclude that midbrain dopamine can provide a teaching signal for initiation of non-Hebbian DA-LTP and contextual memory.

The manuscript is well written. The implementation of the experiment is elegant. The findings will constitute an important contribution to the field of neuronal plasticity. I am thus supportive of this study.

I have one general comment that I think the authors should try to address to further increase the impact of their study. There is an increased interest about the role of dopamine from noradrenergic projections. While the observations reported here are already forming a major contribution to the question of the role of dopamine in LTP, one is to wonder whether the same role would be obtained with dopamine released from the locus coeruleus (LC). Because the authors are able to study LTP in vivo (an important difference compared to past studies using brain slices), this additional data would allow a clear comparison between these two pathways (I am assuming that LC is indeed releasing DA

but this is not my area of expertise).

Minor comments:

Figure 1: a picture showing ventral tegmental area dopamine projections is needed.

Abstract: make clear that this work is about dorsal CA1 (same on line 118 for the conclusion). Likewise, make clear that this LTP study is performed in vivo!

Figure 1b: specify that these are DAT mice; the TH in Figure 1a might lead the reader thinking this work is done in TH mice.

Reviewer #3 (Remarks to the Author):

The manuscript of Sayegh et al. targets to show the role of dopamine in hippocampal synaptic plasticity and context learning. The authors used DAT-Cre mice and optogenetic approaches to manipulate dopamine projections from the ventral tegmental area to hippocampus during in vivo electrophysiological recordings and behavior. The first experiment was conducted in anesthetized mice. It shows that optogenetic activation of hippocampal dopamine terminals concomitant with electrical stimulations of Schaffer collaterals induces long-term potentiation of field potentials. In the second set of experiments, the authors optogenetically manipulated VTA dopamine hippocampal terminals during contextual fear conditioning test. The findings suggest that dopamine activation and inhibition facilitates and attenuates hippocampus dependent learning, respectively. This is an elegant and straightforward study, but there are several issues that require further clarification.

1) The first set of experiments provide novel and compelling evidence for the direct role of hippocampal dopamine in generation of non-hebbian LTP. For the second part, the authors use an elegant behavioral test to show that hippocampal dopamine contributes to contextual fear learning. But this result has already been shown in previously published work (Broussard et al., 2016, and Tsetsenis et al., 2021). Could the authors reiterate the novelty of their behavioral findings and clarify the differences from already published studies?

2) Did the authors measure the effect of dopamine terminals stimulation during 2- and 8-min context pre-exposure? And what is the effect of dopamine terminals inhibition on 30-sec and 8-min context pre-exposure? This information could be important for better understanding of VTA dopamine role in the contextual learning.

3) Fig 2 states that dopamine in the hippocampus is responsible for learning a new context. Fig 2D supports this statement showing that VTA dopamine terminals stimulation during 30 sec of context pre-exposure can enhance learning. But rats are able to learn context with 2-8 min of pre-exposure even without stimulation of DA. Furthermore, inhibition of dopamine terminals in Fig 2E reduces but doesn't block contextual learning. These data suggest that dopamine may contribute but is not "responsible" for learning.

3) What was the rationale for choosing the number of bursts (90) during 30 second context exposure in Fig 2D? This stimulation pattern doesn't match with Fig 1 (1 burst every 30sec), potentially disconnecting the findings from two figures.

4) It is peculiar that the total of 19 ChETA mice was used for the conditioned context in Fig 2D. But I can only see 16 data points in the graph for the alternative context. What happened with the 3 missing mice?

5) It is also interesting that the YFP group exposed to green light stimulation in Fig 2E shows higher freezing than the control group in Fig 2B (79% vs 63%). Do the authors have any explanation for the observed effect?

6) Figs 1A and 2C should have more information. Are these horizontal or sagittal sections? Where is the VTA and where is the SNr on these figures? What are the values of scale bars? Adding figure captions will be helpful.

Dear Reviewers,

First, we would like to apologize for the long delay in responding. We sincerely express our gratitude to the three of you for your generous acknowledgment of the significance of our work and for providing fair and constructive comments. Your insightful feedback has played a pivotal role in enhancing the overall quality of our manuscript entitled "Ventral Tegmental Area dopamine projections to the hippocampus trigger long term potentiation and contextual learning".

Here, we present an improved version of our manuscript that incorporates substantial revisions, addressing the important concerns raised by the three reviewers. As Reviewer 1 correctly surmised, this is a resubmission of a manuscript that was initially formatted as a brief communication. We appreciate your guidance on areas requiring additional development. The manuscript has undergone extensive rewriting, and a new figure has been added to comprehensively address the concerns raised by the reviewers. New Figure 1 now focuses on histology, featuring a set of pictures illustrating the fibers in the hippocampus, while Figure 2 delves into electrophysiology, and Figure 3 explores behavior. The modifications to the manuscript are highlighted in yellow. Below, you will find—written in blue—a point-by-point response to each of your comments (quoted in *italic*).

We have made every effort to address each of your insightful queries and sincerely hope you will find the new version of the manuscript suitable for publication.

With our best regards,

Fares Sayegh & Lionel Dahan

Reviewer #1 (Remarks to the Author):

In this brief manuscript the authors investigated the pairing of single pulses of Schaffer-collateral electrical stimulation with optogenetic burst stimulation of VTA dopamine terminals, on the strength of CA1 hippocampal fEPSPs. They investigate (a) the effect of varying the relative timing of activation of these inputs, (b) the number of pairings of afferent activation with dopamine release, (c) the effect of manipulating dopamine D1-receptor activation on plasticity, and (d) the behavioral effect of DA burst firing or inhibition at the stage of learning a context, prior to learning foot-shock association in that context.

The strengths of this study are: (i) that it is in vivo, (the use of anesthesia is always an issue that most slice physiologists push back on, however I am not one!), (ii) that the authors show convincing immunohistochemistry that demonstrates sufficient and specific transduction of DA cells in DAT-Cre mice, (iii) they use appropriate inactive laser and control-vector controls, (iv) they attempt to clarify the importance of relative timing of afferent inputs to plasticity effects, and (iv) they undertake separate behavioral experiments that aim to give 'context' to the plasticity effects, by demonstrating that dopamine burst stimulation in awake behaving mice enhances contextual learning (by increasing freezing behavior to a previous foot-shock-paired environment), whereas inhibiting dopamine firing decreases contextual learning to control levels.

For the most part, the study is well-performed and convincing, but the weaknesses include (i) lack of detail of some methodological steps, (ii) issues with interpretation of some of the

experiments, and (iii) the degree of novelty of the study in addressing the role of dopamine in hippocampal plasticity induction. The authors should consider the following:

Thank you for acknowledging the robustness of our experimental design and our experimental results. As you correctly surmised, this is a resubmission of a manuscript that was initially formatted as a brief communication. We now better introduce the rationale of the study and discuss the novelty and consequences of our work.

1) The introductory paragraph gives a very abbreviated description of the background to hippocampal LTP (the m/s was I assume submitted as a brief communication but this was not clear from the website which classifies it as an "Article" – even so, the background and justification should be a more thorough). I am sure the authors are aware that there are many forms of induction protocol for hippocampal LTP, many that have had extensive work detailing the cellular induction mechanisms. This background needs expansion, summarising briefly the different forms of induction protocol and focussing this down onto the particular form for which this paper seeks to reveal a novel mechanism.

We appreciate your advice regarding the parts that needed additional development. The introduction has been extended to provide a better context for the study. The many forms of the induction protocol are detailed in lines 25 to 32, and the details of the mechanisms for their initiation and maintenance are explained in lines 34 to 47.

2) The plasticity induction protocols used for each experiment are buried within the text of the legends and only briefly described – there needs a figure inset illustrating exactly the pairing protocol used and the relative pairing timings used in experiments that varied these. What intensity was the electrical stimulation set to that was applied during the plasticity-induction protocol (I realise that fEPSPs were elicited at 70% max response but how was this varied during the induction protocol and what was the post-synaptic effect)? Was there evidence of population spike induction during the protocol or was the intensity subthreshold? Was there a fibre volley visible and if so how did it change? What was the interval between each of the 50 pairings (i.e. interval from electrical-laser to electrical-laser stimulation)?

We completely agree about the lack of methodological information. Actually, we had written the methods in the supplemental material in order to format our previous manuscript as a brief communication. In this new version as a full article, we put the methods back into the manuscript itself (lines 389 to 520) and added a figure inset illustrating the pairing protocol in what is now figure 2a.

The methods section now specifies that the intensity of the stimulation was set to elicit 70% of the maximal fEPSP response throughout the experiment, including the plasticity-induction protocol (lines 413 to 415). There was no increase in the electrical stimulation protocol during the pairings. The stimulating intensity was below the threshold for reliably detecting any population spike in most of our recordings. This is now mentioned in the discussion line 270 and 271. We did not observe any fiber volley in our experiments. Finally, the interval between two consecutive electrical-light pairings was set to 30 seconds, with a jitter of 5 seconds (interval between 27.5 to 32.5 seconds). This is mentioned in the methods (lines 408 to 410) and discussion (lines 264 to 267).

3) *What was the detail of the optogenetic activation of dopamine fibres that was unpaired with Schaffer collateral stimulation? Were the same number of Schaffer-collateral stimulations given but randomised in relation to the optogenetic activations or was the electrical stimulation not given at all? Was the optogenetic activation given as the same number of activations over a similar total period as the paired experiments (assuming that these applications were randomised in comparison to electrical stimulations). This is a very important experiment since if the same number of stims as opto activations are given but one of them is randomly applied in relation to the other it would speak more to the plasticity induced during pairing still as a Hebbian mechanism with the addition of a temporally-specific third factor (dopamine) as a neuromodulator (see Izhikevich, E.M. Solving the distal reward problem through linkage of STDP and dopamine signaling), rather than, as the authors refer to it, a non-Hebbian induction protocol.*

The unpaired group received the exact same number of Schaffer-collateral stimulations and the exact same number of optogenetic stimulations as the paired group. The only differences being that, in the unpaired group, there was a delay of 15+/-2.5 seconds between the optical and the electrical stimulation, while both stimulations were concomitant in the paired group. This is now stated in the results section (lines 167-170).

We fully concur with the reviewer that the term “neoHebbian LTP” is more appropriate than “non-Hebbian LTP”. We intended to convey precisely that, in our conditions, dopamine fulfills the criterion for a third factor, as proposed by Gerstner et al. 2018 or Lisman, Grace & Duzel 2011. Therefore, we have changed the term to “neoHebbian” throughout the manuscript and this interpretation of our data is now presented in the discussion, lines 294 to 299 and 321 to 323.

4) *The justification that the authors give for the novelty of their study states that “no study addressed whether and how midbrain dopamine may trigger hippocampal LTP[14], which is the missing part for fully understanding dopamine's role in allowing supervised learning outside the Hebbian framework[15]”. I would challenge this with the findings of Brzosko, Z. et al (2015), who showed that a post-pre pairing at a very short interval (-10ms) was able to induce LTP due to release of endogenous dopamine. Their plasticity and its timing dependence was able to be blocked by bath-applied D1 antagonist SCH23390. Hence, although the present study used HFS and measured fEPSP slope, there still would have been a conjunction of pre- and post-activity coupled with a ‘third factor’ involving dopamine release working at D1 receptors. This study should be considered in the context of the current findings as evidence of DA D1 involvement in the induction of hippocampal plasticity.*

We acknowledge that this justification was not the best we could have provided. Once again, this was a consequence of adapting our paper to the brief communication format. We now provide in the introduction an in-depth description of what is known regarding the involvement of dopamine in hippocampal synaptic plasticity, including references to Brzosko 2015, among others (lines 72 to 80 and 95 to 109) and a more precise and rigorous justification for the novelty and aim of our study (lines 80 to 83 and 109 to 113 and 115 to 120).

Please note that, except for the TBS in the occlusion experiment, we did not use any HFS. Instead, we simply paired a single-pulse stimulation of Schaffer collaterals repetitively with a burst of dopamine (we have tried to make this clear in the results, lines 263 to 267, and methods sections, lines 413 to 415). We cannot with certainty rule out that the postsynaptic elements may have responded to the stimulation of the

presynaptic elements by firing action potentials, like in Brzosko experiment. However, in the absence of optogenetic dopamine stimulation, electrical stimulation of the Schaffer collaterals failed to trigger any plasticity, contrary to what Hebb's postulate would have predicted if the stimulation of the presynaptic elements were to repeatedly induce postsynaptic action potentials (see discussion lines 268 to 280). This is also different from the STDP protocol used in Brzosko et al., 2015, which elicited a synaptic depression in the absence of dopamine. We therefore agree that the third factor hypothesis correspond to our data, but in a way that is different from Brzosko study (see discussion lines 286 to 299).

5) Fig. 1e is given as evidence that plasticity induced by their DA-LTP pairing protocol occludes subsequent plasticity induction by another protocol TBS, suggesting a common mechanism. Notably, there was an induction of DA-LTP first, then a readjustment of stimulus intensities to baseline (pre potentiation) levels and then the application of TBS. They describe the result as a partial occlusion, however the immediate magnitude of potentiation induced by TBS maps on exactly to control levels; it is just the persistence which is reduced, albeit a change still there in an hour. A true occlusion of mechanism should not have produced any further increase in fEPSP after baseline levels were readjusted. I think this is weak evidence that should be enhanced or qualified with further experiments/analyses or substituted with another experiment seeking mechanism.

The occlusion experiment was lacking some context and explanations which are now provided in the results section, lines 174 to 177. Both experimental groups exhibited similar levels of post-tetanic potentiation, a short-term plasticity typically induced by TBS and independent of LTP mechanisms. Conversely, the subsequent LTP was maintained at a lower magnitude in the DA-LTP group ($23.9\pm 5\%$) compared to the No DA-LTP group ($47.2\pm 6\%$) (figure 2d). Therefore, DA-LTP partially occluded LTP triggered by TBS. This is now explained in the results section (lines 177 to 185).

We agree that this is only partial occlusion, but it is similar to other partial occlusion experiments such as found for instance in Whitlock et al. Science 2006. We however acknowledge that this not a perfect demonstration of a common mechanism and toned down the conclusion of the occlusion experiment with a more cautious statement: "(...) we suspect the presence of a shared underlying mechanism governing the expression and maintenance of DA- LTP and TBS-LTP." (lines 185 to 187).

6) Minor – there are quite a lot of typos/tense and plural issues that will need correcting.

In light of this realization, we asked a native speaker (Peter Redgrave) to proofread our revised manuscript to ameliorate its intelligibility.

Reviewer #2 (Remarks to the Author):

The authors demonstrate that dopamine projections from the ventral tegmental area (VTA) can initiate long term potentiation (LTP) in the dorsal hippocampus of anesthetized mice. This is shown using repeated in vivo electrical stimulation of the Schaffer collaterals with optogenetic stimulation of midbrain dopamine afferents to CA1. The successful induction of hippocampal DA-LTP involves D1/5 receptors and enables mice to acquire contextual fear conditioning. Authors conclude that midbrain dopamine can provide a teaching signal for initiation of non-Hebbian DA-LTP and contextual memory.

The manuscript is well written. The implementation of the experiment is elegant. The findings will constitute an important contribution to the field of neuronal plasticity. I am thus supportive of this study.

I have one general comment that I think the authors should try to address to further increase the impact of their study. There is an increased interest about the role of dopamine from noradrenergic projections. While the observations reported here are already forming a major contribution to the question of the role of dopamine in LTP, one is to wonder whether the same role would be obtained with dopamine released from the locus coeruleus (LC). Because the authors are able to study LTP in vivo (an important difference compared to past studies using brain slices), this additional data would allow a clear comparison between these two pathways (I am assuming that LC is indeed releasing DA but this is not my area of expertise).

We agree that there is a possibility that LC noradrenergic and/or dopaminergic neurons may also trigger hippocampal LTP and memory, and that it would be an important question to address. But we consider it would constitute a completely different work, which is already well advanced by the labs led by John Dany (Tsetsenis, Front Cell Neurosci. 2022) and Richard Morris (see Tse et al., PNAS 2023). We now present the data concerning the role of the LC in the introduction to define more precisely the aim of our study (lines 85 to 93) and discuss this important question in detail in the new version of the discussion (lines 245 to 256 and 346 to 376).

Minor comments:

Figure 1: a picture showing ventral tegmental area dopamine projections is needed.

We added the required picture. Since it made the others figures too crowded, we gathered all immunohistochemistry panels in one figure (figure 1 of the new manuscript).

Abstract: make clear that this work is about dorsal CA1 (same on line 118 for the conclusion). Likewise, make clear that this LTP study is performed in vivo!

We agree, these two points are now clearly stated in the abstract (line 11), discussion (lines 238 to 240) and the conclusion (line 380 and 381).

Figure 1b: specify that these are DAT mice; the TH in Figure 1a might led the reader thinking this work is done in TH mice.

We agree, and modified this figure (new figure 2a) and the title of the corresponding legend (line 543) to specify that we used the DAT::Cre genotype.

Reviewer #3 (Remarks to the Author):

The manuscript of Sayegh et al. targets to show the role of dopamine in hippocampal synaptic plasticity and context learning. The authors used DAT-Cre mice and optogenetic approaches to manipulate dopamine projections from the ventral tegmental area to hippocampus during in vivo electrophysiological recordings and behavior. The first experiment was conducted in anesthetized mice. It shows that optogenetic activation of hippocampal dopamine terminals concomitant with electrical stimulations of Schaffer collaterals induces long-term potentiation of field potentials. In the second set of experiments, the authors optogenetically manipulated VTA dopamine hippocampal terminals during contextual fear conditioning test. The findings suggest that dopamine activation and inhibition facilitates and attenuates hippocampus

dependent learning, respectively. This is an elegant and straightforward study, but there are several issues that require further clarification.

1) The first set of experiments provide novel and compelling evidence for the direct role of hippocampal dopamine in generation of non-hebbian LTP. For the second part, the authors use an elegant behavioral test to show that hippocampal dopamine contributes to contextual fear learning. But this result have already been shown in previously published work (Broussard et al., 2016, and Tsetsenis et al., 2021). Could the authors reiterate the novelty of their behavioral findings and clarify the differences from already published studies?

Thank you for this very important comment. Once again, this explanation was missing because of space limitation in order to save space for the original submission as a brief communication.

Indeed, Tsetsenis et al., 2021, by using similar optogenetic tools as we use here, showed that increasing dopamine during fear conditioning improves its encoding, consistent with our results. In addition to replicate their findings, we here provide two new pieces of information: 1/ by using the Context Pre-exposure Facilitation Effect (CPFE), we show that dopamine is required for learning the context itself, independently from the fearful stimuli and 2/ we provide the first evidence that inhibiting the dopamine neurons projecting to the hippocampus prevents memory encoding.

We made clarified the novelty of our behavioral findings and the differences from already published studies in the discussion sections (327 to 344)

2) Did the authors measure the effect of dopamine terminals stimulation during 2- and 8-min context pre-exposure? And what is the effect of dopamine terminals inhibition on 30-sec and 8-min context pre-exposure? This information could be important for better understanding of VTA dopamine role in the contextual learning.

We actually choose to increase dopamine in the 30sec preexposure and to inhibit dopamine during the 2 minutes preexposure because this was the more suited experiments to answer the question we are asking, while reducing the number of mice used in our experiments according to ethical guidelines.

The freezing behavior is already quite high following the 2 to 8 minutes pre-exposure protocol (fig 3b in the new manuscript). An activation of the dopamine system might indeed increase the freezing behavior in these conditions but the size of the effect would be limited by a ceiling effect and would require a huge sample size to reach reasonable statistical power.

On the other side, the 30 seconds pre-exposure does not induce any facilitation effect (fig 3b in the new manuscript). Thus, we do not expect inhibiting the dopamine system to have any effect. Conversely, inhibiting dopamine terminals during 8 minutes of pre-exposure might be of interest, however, such a long activation of eNpHR3.0 opsin was show to have nonspecific and damaging effects (Yizhar O. et al. Neuron 2011).

Thus, we believe the reviewer will agree that, although it was tempting to add these additional experimental groups, they are not justified from both scientific and ethical points of view.

3) Fig 2 states that dopamine in the hippocampus is responsible for learning a new context. Fig 2D supports this statement showing that VTA dopamine terminals stimulation during 30 sec of context pre-exposure can enhance learning. But rats are able to learn context with 2-8 min of pre-exposure even without stimulation of DA. Furthermore, inhibition of dopamine terminals in Fig 2E reduces but doesn't block contextual learning. These data suggest that dopamine may contribute but is not "responsible" for learning.

The optogenetic inhibition brings the level of freezing to the level observed in the non pre-exposed group. Thus, even if there is a residual freezing, it does not correspond to the residual learning of the context during the pre-exposure but to the effect of the immediate shock condition. We modified the result section in order to better explain this point (lines 228 to 231).

3) What was the rationale for choosing the number of bursts (90) during 30 second context exposure in Fig 2D? This stimulation pattern doesn't match with Fig 1 (1 burst every 30sec), potentially disconnecting the findings from two figures.

It is challenging to match the optical stimulation protocols between electrophysiology and behavior experiments. In electrophysiology experiments, we precisely control the timing of activation of Schaffer collaterals, which is not the case for the behavior. Tsetsenis et al., 2021 also employed a similar optical burst stimulation of dopamine terminals throughout the fear conditioning protocol. We specifically chose this protocol to align with the activity of dopamine neurons we had previously recorded during REM sleep and palatable food consumption (Dahan et al., 2007). This information is now detailed in the methods section (lines 471 to 472).

4) It is peculiar that the total of 19 ChETA mice was used for the conditioned context in Fig 2D. But I can only see 16 data points in the graph for the alternative context. What happened with the 3 missing mice?

Thank you very much for spotting this. There were actually 3 points at 0% freezing hidden by a mistake when generating the figure. When double checking all behavioral data, we realized there was also one missing point at 0% freezing for the YFP group. Both mistakes are corrected.

5) It is also interesting that the YFP group exposed to green light stimulation in Fig 2E shows higher freezing than the control group in Fig 2B (79% vs 63%). Do the authors have any explanation for the observed effect?

These experiments were done on different batches of mice at different times. Noteworthy, the percentages of freezing in the modified context of both groups in what is now fig 3d (the eNpHR3.0 experiment) are higher than in any other previous experiment. Thus, our favorite hypothesis for this difference is a batch effect on the unspecific tendency of the mice to exhibit freezing behavior.

Since it doesn't affect the conclusion of this experiment, we did not discuss this point in the manuscript. In case the reviewer really feels it is mandatory, we could add a few sentences in the discussion.

6) Figs 1A and 2C should have more information. Are these horizontal or sagittal sections? Where is the VTA and where is the SNr on these figures? What are the values of scale bars?

We totally agree with these comments; Fig 1A and 2C have been merged into Figure 1, which, along with the legend (lines 523 to 538), has been modified in accordance with the reviewer's advice.

REVIEWERS' COMMENTS

Reviewer #1 (Remarks to the Author):

The authors have extensively revised their manuscript in response to the issues raised by myself and the other two reviewers. It is now significantly longer, clearer and more interesting and convincing, with the additional explanations and introduction/discussion given. Although I still am not enamoured by the interpretation of the occlusion experiments I am happy to let this go. The authors are to be congratulated for tackling the question of whether midbrain dopamine cells can interact with Schaffer collaterals to induce potentiation effects in the CA1 of the hippocampus – it now convincingly does. There remains a few minor typos after the extensive English revision, but these will be picked up by proof readers. I enthusiastically endorse this paper.

Reviewer #2 (Remarks to the Author):

The authors have addressed all my comments in full. Well done for having delivered such an important work.

Reviewer #3 (Remarks to the Author):

All my comments were addressed